# A Joint Network of Edge-Aware and Spectral–Spatial Feature Learning for Hyperspectral Image Classification

**DOI:** 10.3390/s24144714

**Published:** 2024-07-20

**Authors:** Jianfeng Zheng, Yu Sun, Yuqi Hao, Senlong Qin, Cuiping Yang, Jing Li, Xiaodong Yu

**Affiliations:** 1College of Computer Science and Information Engineering, Harbin Normal University, Harbin 150025, China; jianfeng_zheng@stu.hrbnu.edu.cn (J.Z.); cathy_hao@stu.hrbnu.edu.cn (Y.H.); qinsenlong@stu.hrbnu.edu.cn (S.Q.); ycphsd@hrbnu.edu.cn (C.Y.); 2Department of Municipal and Environmental Engineering, Heilongjiang Institute of Construction Technology, Harbin 150025, China; donglin_2016@nefu.edu.cn

**Keywords:** hyperspectral image (HSI) classification, edge feature augment, feature extraction

## Abstract

Hyperspectral image (HSI) classification is a vital part of the HSI application field. Since HSIs contain rich spectral information, it is a major challenge to effectively extract deep representation features. In existing methods, although edge data augmentation is used to strengthen the edge representation, a large amount of high-frequency noise is also introduced at the edges. In addition, the importance of different spectra for classification decisions has not been emphasized. Responding to the above challenges, we propose an edge-aware and spectral–spatial feature learning network (ESSN). ESSN contains an edge feature augment block and a spectral–spatial feature extraction block. Firstly, in the edge feature augment block, the edges of the image are sensed, and the edge features of different spectral bands are adaptively strengthened. Then, in the spectral–spatial feature extraction block, the weights of different spectra are adaptively adjusted, and more comprehensive depth representation features are extracted on this basis. Extensive experiments on three publicly available hyperspectral datasets have been conducted, and the experimental results indicate that the proposed method has higher accuracy and immunity to interference compared to state-of-the-art (SOTA) method.

## 1. Introduction

HSIs are complete spectral information for each pixel generated by hyperspectral sensors by capturing the reflection information of an object in multiple consecutive spectral bands through hyperspectral imaging technology. Compared to RGB images, HSIs also contain the information about the shape, texture, and structure of the object [1], but HSIs contain a large amount of waveband information, which allows identification and differentiation of substances with similar colors but different spectral characteristics. Thus, HSIs are widely used in scientific and industrial applications that require precise substance identification and analysis, such as medical imaging and diagnosis [2], geological and mineral exploration [3], environmental protection [4], agricultural crop monitoring [5], food safety monitoring [6], and military reconnaissance and security [7]. To fully exploit the value of HSIs, many subtasks are derived, such as classification [8,9], target detection [10,11,12], and unmixing [13,14,15]. Among these tasks, the land cover information classification task has received extensive attention.

When classifying objects in HSIs, due to the phenomenon that “the spectra of the same object may be different and the spectra of different objects may be the same” exists in HSIs [16], therefore, it is not feasible to simply apply the same methods used for the RGB image classification to the HSIs classification. To address the above challenges, researchers around the world have proposed various approaches, such as principal component analysis (PCA) [17], the Bayesian estimation method [18], SVM [19,20], and k-mean clustering [21,22].

However, with the breakthrough of deep learning, convolutional neural networks (CNNs) are gradually replacing the traditional HSI classification methods due to their stronger model generalization ability and deep feature characterization. And in recent years, in the field of HSI classification, CNNs have been rapidly developed. For example, Hu et al. used 1-D CNN [23] in order to extract the spectral information. But for the HSI classification task, using only spectral information is not enough to realize to obtain accurate classification results. Therefore, Zhao et al. proposed 2-D CNN [24] to extract spatial features. However, both 1-D CNN and 2-D CNN do not fully utilize the 3-D characteristics of HSIs. Thus, Chen et al. applied 3-D CNN [25] to the field of HSI classification in order to fuse the spatial–spectral features of HSIs, and the experimental results showed that the performance of the model was improved. Based on these experiments, many researchers have proposed hybrid convolutional methods [26,27,28,29,30,31,32]. Among them, Roy et al. proposed the HybridSN [29] with a linear structure. HybridSN contains three sequentially connected 3D convolutional layers for fusing spatial and spectral information and one 2D convolutional layer for extracting local spatial features, respectively. In addition, Zhong et al. proposed the SSRN [32], where they introduced residual connectivity between convolutional blocks to promote backpropagation of the gradient. However, these convolution-based methods are limited by the convolutional kernel, which can only learn the information within the coverage of the convolutional kernel, thus restricting the representation of global features.

During this period, Dosovitskiy et al. proposed the vision transformer (ViT) [33], and the multi-head attention mechanism proposed in the paper greatly alleviated the problem of sensory field limitation of convolution-based methods. Since then, a large number of methods combining ViT and CNN appeared [33,34,35,36,37,38,39,40,41,42,43,44,45], such as SSTN [45], SSFTT [37], CTMixer [43], and 3DCT [39]. These methods have both global feature perception capability based on ViT methods and local feature fusion capability based on CNN methods. Thus, compared to methods [46,47,48,49] that only use CNN to obtain spectral–spatial features, these mixed-use CNN and ViT methods are able to extract more comprehensive spectral–spatial features due to the use of a global attention mechanism.

However, the characteristics of HSIs, especially the importance of edge features between different classes and spectral bands in the classification process, are not fully considered in these methods that use a mixture of CNN and ViT. To enhance edge features, edge data augmentation methods are often employed. Traditional image edge data augmentation methods usually apply edge detection operators (e.g., Laplacian, Canny, Sobel operators) [50] directly on the original image to obtain the edge information, which is then used for subsequent model training by directly superimposing with the original image. However, in the field of HSI classification, due to the existence of the characteristic that the boundaries of the same object may be different in different spectral bands, processing the original data in this way will cause a large amount of noise, which will affect the subsequent classification performance. In order to minimize the effects of superimposed noise, Tu et al. applied edge-preserving filtering to the edge portion in their proposed MSFE [51] with pyramidal structure, but the MSFE does not take into account the fact that different spectral bands play different roles in the classification process.

Therefore, inspired by the above work, and in order to enhance the image features and weaken the noise interference of the initial HSI, we adopt a dynamic learning approach to obtain the edge information and the decision weights of different spectral bands. Then, we use a mixture of attention mechanisms and CNN on this basis with the aim of obtaining global spectral–spatial features. Figure 1 shows the edge-aware and spectral–spatial feature extraction network which we propose. The network contains two parts: an edge feature augment block and a spectral–spatial feature extraction block. Different from traditional data augmentation that is not dynamically learnable, our edge feature augment block adaptively learns the degree of edge feature enhancement in different spectral bands, which reduces the high-frequency noise. In addition, in the spectral attention block, we adaptively adjust the weights of different spectral bands for classification and then perform feature extraction on its basis. To sum up, there are three main contributions:We propose a novel feature extraction network (ESSN) with richer and more efficient representation of edge features and spectral–spatial features compared to existing networks;We designed a novel edge feature augment block. The block consists of an edge-aware part and a dynamic adjustment part. Compared with edge data augmentation methods that are not dynamically learnable, this block greatly reduces edge distortion and noise amplification;We propose a spectral–spatial features extraction block. It contains a spectral attention block, a spatial attention block, and a 3D–2D hybrid convolution block. Among them, the spectral attention block and the spatial attention block gain an effective feature by enhancing the information favorable for classification and suppressing noise and other interfering information. The convolution block fuses the above features.

The subsequent sections are composed as follows. Our proposed method is described in Section 2. In Section 3, we describe our experimental environment and make a detailed comparison with other SOTA methods in the same environment. We perform sensitivity analysis experiments and ablation experiments aimed at verifying the importance of each part of the model in Section 4. In Section 5, we distill the paper and suggest directions for model improvement.

## 2. Methodology

Figure 1 shows the whole process of HSI classification. It consists of a data preprocessing block, the backbone of the proposed network, and a linear classifier.

Real objects are given a hyperspectral image (HSI) after passing through a hyperspectral sensor. Assuming that the HSI is $I_{raw}\in R^{H\times W\times C}$. H, W, C are the height, width, and number of spectral bands of the raw HSI image, respectively. In HSI, each pixel can be represented by the vector $X_{pixel}=(V_{1},V_{2},\ldots ,V_{c})$, where $V_{c}$ represents the pixel value on the *C*th spectrum. Obviously, the greater the number of spectra, the richer the information, but this greatly slows down computational efficiency. Therefore, we adopt the PCA technique to preprocess the HSI data to improve the efficiency, maintain the height and width unchanged, and reduce the spectral number from C to P. We denote the HSI after PCA dimensionality reduction as $I_{pca}\in R^{H\times W\times P}$, where P denotes the number of the spectra after PCA dimensionality reduction. In order to obtain a suitable input format for the network, we crop the image into patches $I_{patch}\in R^{h\times w\times P}$ with pixel-centered dimensions as h, w, P, where h, w, P represent the height, width, and spectral number of the patch, respectively. The data preprocessing block is shown in Figure 1. Note that the same symbols appearing in this section represent the same meaning.

The backbone of ESSN contains both an edge feature augment block and a global spectral–spatial feature extraction block, and we will describe the content of ESSN in as much detail as possible. Finally, we use a linear classifier to determine the class of each pixel.

### 2.1. Edge Feature Augment Block

As shown in Figure 2, the points where the model fails to predict are mostly at the intersection of different categories, which is due to the high feature similarity between certain categories on the one hand, and the possible boundary blurring between certain categories [52].

Previously, edge data augmentations were usually used to strengthen the edge for the above problems. However, the direct superposition of edge information may produce strong edge noise, leading to the confusion of similar categories. Therefore, we propose a novel edge feature augment block, as shown in Figure 1, which can adaptively adjust the model’s emphasis on the edges of a region by learning the importance of the edge information in the input data, and personalize the edge information.

#### Laplacian of Gaussian Operator

The Laplacian of Gaussian operator is generated by the convolution of the Laplace operator and Gaussian filtering operator. The Laplace operator is particularly sensitive to regions of the image that change abruptly, and therefore has a better performance in edge-awareness tasks. Due to the prevalence of Gaussian noise in images captured by electronic devices, which seriously affects the accuracy of edge perception, hyperspectral images need to be processed with Gaussian filtering before perceiving the edges. The Gaussian filtering operator and Laplace operator can be expressed by Equations (1) and (2), respectively:(1)$$G_{\sigma }\left(x,y\right)=\frac{1}{2\pi \sigma^{2}}e^{\left(-\frac{x^{2}+y^{2}}{2\sigma^{2}}\right)}$$
(2)$$L\left(x,y\right)=\frac{\partial^{2}I}{\partial x^{2}}+\frac{\partial^{2}I}{\partial y^{2}}$$
where x,y denote the spatial coordinate positions of the HSIs, σ is the Gaussian standard deviation, and I represents the value of the pixel on the image.

Convolutional operations have the law of union, so we use the result of the convolution of the Gaussian filter operator with the Laplace operator as a new edge-aware operator (LoG) and then use the LoG to convolve the image to obtain the image edges. The LoG expression is shown in Equation (3).
(3)$$LoG\left(x,y\right)=-\frac{1}{\pi \sigma^{4}}[1-\frac{x^{2}+y^{2}}{2\sigma^{2}}]e^{\left(-\frac{x^{2}+y^{2}}{2\sigma^{2}}\right)}$$

Due to the discrete representation of hyperspectral images, we discretize Equation (3) to obtain an approximate LoG operator for practical use. As shown in Figure 3, we list the LoG operators for the two cases σ<0.5 and σ=1.4, respectively. Then, let the result after edge-aware operator be $I_{LoG}$ with the following expression:(4)$$I_{LoG}=DWConvLoG(I_{patch})\in R^{h\times w\times P}$$
where DWConvLoG(⋅) indicates depthwise separable convolution with the kernel of LoG.

In the edge feature augment block, because of the characteristic that “the spectra of the same object may be different and the spectra of different objects may be the same”, strengthening the edge features at the same rate in different spectral bands will generate interference noise, so we design a learnable parameter $\gamma \in R^{1\times P}$ for adjusting the degree of feature augment in different spectra. We explore the importance of γ in Section 4.2. And in order to make the network more flexible and the optimization process smoother and more efficient, we use residual connectivity. The output ${(I}_{LoGout})$ of the module is shown below:(5)$$I_{LoGout}=I_{patch}+\gamma ⨂(\varphi (I_{LoG})⨂I_{patch})\in R^{h\times w\times P}$$
where φ(⋅) indicates activation function of sigmoid. ⨂ denotes the dot product of the corresponding position.

### 2.2. Spectral–Spatial Feature Extraction Block

#### 2.2.1. Spectral Attention Block

HSIs are rich in spectral information; to make it easier to see, we show the specific image of each spectral band by means of a grayscale map, as shown in Figure 4. Obviously, the importance of different spectra in the decision-making process is different [53]. The spectral attention helps the model in adaptive adjustment of weights for different spectra and in enhancing the representation of these spectra during the learning process. This helps the model to suppress the influence of task-irrelevant spectra.

In the spectral attention block, to strengthen the correlation between encoded and decoded data, we use residual concatenation. Let the input features of the block be $I_{input}\in R^{h\times w\times P}$, then the results after global maximal pooling and global mean pooling are $v_{max}$ and $v_{mean}$, respectively, where global maximum pooling is complementary to global average pooling. The results are shown below.
(6)$$v_{max}(p)=Max(I_{input}(p))\in R^{1\times P}$$
(7)$$v_{mean}(p)=\frac{1}{h\times w}\sum_{x=1}^{h}\sum_{y=1}^{w}I_{input}(x,y,p)\in R^{1\times P}$$

To reduce the size of parameters, the pooled features are entered into a shared multi-layer perception (MLP), and the results $h_{max}$ and $h_{mean}$ are obtained. Let the rate of dimensionality reduction be r and the weights of the two MLP layers be, in order, $W_{1}\in R^{p\times (p/r)}$ and $W_{2}\in R^{(p/r)\times p}$. $h_{max}$ and $h_{mean}$ are as follows:(8)$$h_{max}=W_{2}(W_{1}(v_{max}))\in R^{1\times P}$$
(9)$$h_{mean}=W_{2}(W_{1}(v_{mean}))\in R^{1\times P}$$

Adaptive spectral weights $W_{h}$ of the input feature map are obtained by adding $h_{max}$ and $h_{mean}$ and passing through the sigmoid activation function. $W_{h}$ is shown below:(10)$$W_{h}={\sigma (h}_{max}+h_{mean})\in R^{1\times P}$$
where σ(·) is the activation function of sigmoid.

Finally, let the output of the spectral attention block be $I_{output}\in R^{h\times w\times P}$, the expression is as follows:(11)$$I_{output}=I_{input}+I_{input}⨂W_{h}\in R^{h\times w\times P}$$

#### 2.2.2. Spatial Attention Block

In contrast to traditional convolutional operations, which focus on only a portion of the input data, the spatial attention mechanism [54] can adaptively adjust the area of attention over the global spatial range of the input data and give more importance and weight to these locations during preprocessing, thus improving the recognition accuracy and efficiency of the model.

In Figure 5, the structure of spatial attention in the spatial attention block is illustrated. Considering that the spatial information of the same location may behave differently on different spectral bands, we first fused the local spatial features by 2D convolution and then projected the convolved feature maps to obtain Q, K, and V, respectively:(12)$$Q=Conv2D(I)W_{Q}\in R^{hw\times p}$$
(13)$$K=Conv2D(I)W_{K}\in R^{hw\times p}$$
(14)$$V=Conv2D(I)W_{V}\in R^{hw\times P}$$Then, the attention map Attn can be calculated as follows:(15)$$Attn=Softmax(Q\cdot K^{T}/\sqrt{d_{k}})\in R^{hw\times hw}$$
where $d_{k}$ is the dimension of K. Let ${Output}_{sa}$ be the output in the network of Figure 4, the expression is as follows:(16)$${Output}_{sa}=Attn\cdot V=Softmax(Q\cdot K^{T}/\sqrt{d_{k}})\cdot V\in R^{hw\times P}$$Finally, we reshape ${Output}_{sa}$ to the size of h×w×P for subsequent processing.

In Figure 1, the spatial attention block contains two spatial attention parts; we use two convolutional kernels of different sizes on two spatial attention parts to enhance the perceptual region. In addition, in order to strengthen spatial expression, we use a residual structure.

#### 2.2.3. 2D–3D Convolution

2D convolution layer can extract spatial features, and 3D convolution layer can extract spectral features. Therefore, as shown in Figure 1, 2D convolution and 3D convolution are used in the spectral–spatial feature extraction block. In the 2D–3D convolution block, three consecutive 3D convolutional layers with a different kernel and one 2D convolutional layer are included in the 2D–3D convolutional block. A detailed description is shown below.

In the 3D convolution layer, a single 3D convolution can be regarded as a 3D convolution kernel sliding along the three dimensions (H, W, C). During the convolution process, the spatial and spectral information of the neighboring spectra are fused. And the values of the *n*th feature map of the *m*th layer at the spatial location of (x,y,z) are as follows:(17)$$v_{m,n}^{i,j,k}=\varphi (b_{m,n}+\sum_{\tau =1}^{d_{l-1}}\sum_{\lambda =-h_{m}}^{c_{m}}\sum_{\rho ={-w}_{m}}^{h_{m}}\sum_{\sigma =-c_{m}}^{w_{m}}\omega_{m,n,\tau }^{\sigma ,\rho ,\lambda }\times v_{m-1,\tau }^{i+\sigma ,j+\rho ,k+\lambda })$$
where φ(·) is the activation function, and $b_{m,n}$ and $\omega_{m,n}$ are the bias parameters and weight values of kernel corresponding to the *n*th feature map of the *m*th layer, respectively. $d_{l-1}$ indicates the number of feature maps in the (*l* − 1)th layer and the depth of $\omega_{m,n}$. The height, width, and spectral dimension of the kernel are $\left(2h_{m}+1\right)$, $\left(2w_{m}+1\right)$, and $\left(2c_{m}+1\right)$, respectively.

In the 2D convolution layer, the convolution kernel slides over the entire space, and the output of the convolution is the sum of the dot products between the kernel and the input data. During the convolution process, the information of different spectra in the same space is fully integrated. In 2D convolution, the values of the *n*th feature map of the *m*th layer at the spatial location of (x,y,z) are as follows:(18)$$v_{m,n}^{i,j}=\varphi (b_{m,n}+\sum_{\tau =1}^{d_{l-1}}\sum_{\rho ={-w}_{m}}^{h_{m}}\sum_{\sigma =-c_{m}}^{w_{m}}\omega_{m,n,\tau }^{\sigma ,\rho }\times v_{m-1,\tau }^{i+\sigma ,j+\rho })$$
the parameters appearing in Equation (18) represent the same meaning as in Equation (17).

## 3. Comparison Experiments

### 3.1. HSI Datasets

We apply the model to three public datasets: the Indian Pines (IP), the Kennedy Space Center (KSC), and the Pavia University (PU) to evaluate the validity of model.

The IP size is 145 × 145 pixels, and the spatial resolution is 20 m. In the experiment, noisy bands and water absorption have been removed, and 200 spectral bands have been filtered out. The false-color map, ground truth map, training ratios, and 16 vegetation classes of the IP dataset are shown in Figure 6.

The KSC size is 512 × 614 pixels, and the spatial resolution is 18 m. In the experiment, noisy bands and water absorption have been removed, and 176 spectral bands have been filtered out. The false-color map, ground truth map, training ratios, and 13 wetland categories of the KSC dataset are shown in Figure 7.

The PU size is 610 × 340 pixels, and the spatial resolution is 1.3 m. In the experiment, noisy bands have been removed, and 103 spectral bands have been filtered out. The false-color map, ground truth map, training ratio, and 9 urban land cover categories of the PU dataset are shown in Figure 8.

In order to avoid the effect of dataset randomness as much as possible, all experiments under this section are trained and tested on ten randomly generated identical training sets and corresponding test sets.

### 3.2. Experimental Setting

#### 3.2.1. Measurement Indicators

The overall accuracy (OA), average accuracy (AA), and kappa coefficient (κ) metrics are being used for quantitative evaluation to fairly consider ESSN against other comparative methods. The larger the value of the model corresponding to the above three metrics, the better the model performs.

#### 3.2.2. Environment Configuration

The software environment for all experiments is PyTorch1.12.0, cuDNN8.0, CUDA11.7, and Python3.8. The hardware environment is an Intel i5-12490F CPU, an NVIDIA GeForce RTX 3060 GPU, RAM 32 GB, and 1 TB of memory. The Stochastic Gradient Descent (SGD) optimizer was chosen as the initial optimizer for all experiments, the cross-entropy loss function is used to calculate the loss, the learning rate is set to 0.01, the batch size is 100, and the patch size is 15 × 15. One hundred epochs are applied to each dataset.

### 3.3. Comparison Experiment Result

We have experimented fully in the same environment and compared it with other SOTA models. In total, eight comparison models were selected, including 2-D CNN [24], 3-D CNN [25], HybridSN [29], SSRN [32], SSTN [45], SSFTT [37], CTMixer [43], and 3DCT [39]. In order to attain a relatively fair comparison result, only the model was changed during the comparison process, and the other parts were kept unchanged, so that the influence of other external factors could be minimized.

Table 1, Table 2 and Table 3 show the mean classification results and standard deviation for each category on the IP, KSC, and PU datasets, respectively, as well as the evaluation metrics for each model. Results are in percentage terms. Figure 9, Figure 10 and Figure 11 show the performance of each model on the same training set on the IP, KSC, and PU datasets, respectively. And in Figure 9, Figure 10 and Figure 11, the background is labeled as black pixels, the category to be predicted is labeled as colored pixels.

By looking at the classification result plots, it can be found that the points of classification errors of each model occurred mostly in the regions where multiple categories appeared densely, so we zoomed in on the localized multi-category regions.

#### 3.3.1. Comparative Results on the IP

By observing Table 1, it shows that ESSN outperforms other methods, and the overall results are OA (98.93%), AA (96.49%), and Kappa (98.78%). And compared with the second-best model (HyBridSN [29]), all performance metrics are improved, 0.21% higher in OA, 0.68% higher in AA, and 0.24% higher in Kappa, respectively. This indicates that the inclusion of edge feature augment and the spectral–spatial attention block is beneficial to the feature representation of the model. In addition, it is observed that all models perform poorly on classes ‘1’ and ‘9’, and by analyzing the IP dataset, it can be found that the overall number of classes ‘1’ and ‘9’ is too small. Consequently, when using the approach of proportional random selection for the training dataset, an inadequate number of samples is obtained for this particular class, resulting in suboptimal classification performance by all models for that class.

In addition, from the overall plot in Figure 9, ESSN has the best overall performance, and from the local plots with high category density, ESSN also outperforms the other models in places with high category density. This indirectly shows that the edge feature enhancement is effective. Also, Figure 9 verifies that most of the places where the prediction is wrong are common areas with different categories.

#### 3.3.2. Comparative Results on the KSC

By observing Table 2, it shows that ESSN is higher than the others. The overall results are OA (99.18%), AA (98.79%), and Kappa (99.08%). Compared with the second-best model (HyBridSN [29]), all performance metrics are improved, 0.19% higher in OA, 0.32% higher in AA, and 0.21% higher in Kappa, respectively. In addition, it can be found that ESSN performance is optimal in most categories, and on a small number of categories, the classification results of our model do not differ much from the best classification results.

Based on Figure 10, it can be seen that ESSN outperforms the other models on the KSC dataset in general. As can be seen in the local zoomed-in image, at the edges of the categories in the category-dense region, ESSN’s performance is also better than the other models. The reason for obtaining the above results is that in KSC, in general, each category is more independent, and there are not many sample points that are in close proximity to other categories. The performance of ESSN on the edges is much sharper than the method without edge feature augment.

#### 3.3.3. Comparative Results on the PU

By observing Table 3, the overall performance of ESSN is OA (99.45%), AA (98.82%), and Kappa (99.27%). Compared with the second-best model (3DCT [39]), all performance metrics are improved, where OA is improved by 0.09%, AA by 0.14%, and Kappa by 0.12%, respectively. Combining Table 3 and Figure 7 and Figure 11, compared with the HybridSN [29], which has the highest number of optimal classification accuracies in a single category, it can be found that ESSN’s classification accuracies are competitive, if not optimal, in categories ‘1’, ‘2’, ‘7’, and ‘8’, but ESSN outperforms it significantly in categories ‘3’, ‘4’, and ‘9’. The same result can be observed in the localized zoomed-in plots in Figure 11. This explains why the average classification accuracy of ESSN is higher than that of other methods, even though ESSN’s single-class classification accuracy is not optimal on most categories.

### 3.4. Depletion of Resources

We take the parameter size, training time, and testing time as resource consumption training metrics, and smaller values are better for all indicators. All results are stored in Table 4.

Based on Table 4, compared with 3DCT, although ESSN contains the largest number of parameters, ESSN has a shorter training time. And by comparing SSTN and 2-D CNN, it can be found that the parameter size of the model does not necessarily have a linear relationship with the training time.

## 4. Discussion and Analysis

### 4.1. Parametric Analysis

In this section, we perform a sensitivity analysis on three parameters: patch size, training ratio, and operator of LoG, respectively, and explore their impact on model performance.

In Table 5, ESSN performs better on IP and PU when patch size is selected as 15 × 15 but does not perform as well on KSC as when patch size is 19 × 19. Considering the amount of computation, patch size 15 × 15 is selected as the optimal size. In addition, as the patch size increases, the OA on the KSC becomes larger and larger, and combined with the full ground truth map of KSC, there are two reasons for this result: one is that as the patches increase, each patch contains more spatial information, and thus the model can learn more key elements from it. The other is that as the patches increase, the longer distance edges are gradually incorporated into the model’s observation range. In addition, by looking at the ground truth plots of IP and PU, it can be seen that there is more category intermingling in these two datasets. Increasing the patch can obtain a larger perceptual field which is beneficial to the model but, at the same time, will introduce more noise and confusing information which is detrimental to the model. Therefore, ESSN shows a performance on both IP and KSC that first increases with increasing patch and then decreases with increasing patch. From Table 5, the point of patch size of 15 × 15 is the cutoff point where the model performance goes from up to down.

As seen in Figure 12, apparently, as expected, the performance of all models improves with increasing training samples, with OA gradually approaching 100%. In addition, ESSN has a large advantage when the number of training samples is insufficient, and the OA of ESSN gradually decreases with the increase in samples used for training, and eventually the performance of all models gradually converges to the same. All in all, ESSN outperforms other models on different training ratios.

Figure 13 shows the performance on different cases. Comparing ‘a’ with ‘c’, it is clear that if traditional data enhancement methods are used without processing the raw data with learnable parameters, the performance is not as good as when using edge feature augment blocks. In addition, comparing ‘a’ with ‘e’, it can be found that traditional data augment does not have a positive effect. Especially on the KSC dataset, it greatly reduces its classification performance. When using the edge feature augment block, it can be seen that the performance of the different operators of LoG is very close. And comparing ‘e’ with ‘b’, ‘c’, ‘d’, the classification capabilities of the model are improved when the edge feature augment block is added. Comparing ‘b’ with ‘c’, the performance gap between the two on all the datasets used in the experiment is super small. The reason is that the LoG operators used for both ‘b’ and ‘c’ are discrete approximations at σ<0.5, and the difference between the two is the difference in the angle at which the rotational invariance is satisfied, with the LoG operator corresponding to ’b’ having invariant results for rotations in the 90° direction, and the LoG operator corresponding to ‘c’ having invariant results for rotations in the 45° direction. In this study, raw data are not rotationally transformed, so the difference between the two is not significant, and both are better than the case corresponding to ’e’. Obviously, ‘b’ performs optimally on the KSC, and ‘c’ performs optimally on the IP and PU. After comprehensive consideration, the LoG operator corresponding to ‘c’ is chosen.

### 4.2. Ablation Experiment

In this section, for the influence of external factors, the training samples in the ablation experiments are kept the same as those in the experiments in Section 3. Thus, effects arising from hyperparameters and randomized training samples are excluded.

PCA operation is used in the data preprocessing part, but it also causes loss of spectral information when extracting the principal components of hyperspectral images. Therefore, we explore the effect of PCA operation on the comprehensive performance of the model by conducting ablation experiments of PCA operation on three datasets: IP, KSC, and PU. The experimental results are shown in Figure 14. When PCA operation is used to downscale to 50 dimensions, it can be clearly observed from subplot (a) in Figure 14 that the classification accuracy of the model under PCA operations is not very different from the classification performance of the model without PCA operations, but from subplot (b) in Figure 14, it can be found that the time cost of the model can be greatly reduced by using PCA operations. With the comprehensive consideration of model classification accuracy and time cost, we adopt PCA to preprocess the original data.

In addition, in order to analyze the effect of each component in the model on the model performance, a total of seven combinations were considered in this experiment. All the results are shown in Table 6, which depicts the edge feature augment block with “Edge Block” instead, the spectral attention block with “Spectral Block” instead, and the spatial attention block with “Spatial Block” instead, respectively.

Comparing ‘7’ with ‘1‘ to ‘6’, when all blocks are employed, the model has the best performance.

Comparing ‘7’ to ‘6’, ‘5’ to ‘3’, and ‘4‘to ‘2’, respectively, it can be seen that adding the edge feature augment block improves the model’s performance in IP, KSC, and PU. Combined with the experimental conclusions about edges drawn in Section 4.1, it can be found that the edge feature augment block is different from the traditional edge data augmentation and plays a positive role in improving the network performance. Comparing ‘7’ to ‘5’, ‘6’ to 3, and ‘4’ to ‘1’, respectively, it can be found that the ability to model will be strengthened on IP, KSC, and PU when adding the spectral attention block. Comparing ‘7’ to ‘4’, ‘6’ to ‘2’, and ‘5’ to ‘1’, adding the spectral attention block is also beneficial in improving the model’s performance on IP, KSC, and PU.

Comparing ‘7’ with ‘1’ to ‘6’, it is found that ‘7’ performs optimally on IP, KSC, and PU, which indicates that the combination of edge feature augment block, spectral attention block, and spatial attention block is effective. Then, comparing ‘4’ with ‘1’ and ‘2’, ‘5’ with ‘1’ and ‘3’, and ‘6’ with ‘2’ and ‘3’, it can be found that the performance of the combination of two blocks is better than a single block.

Comparing ‘1’, ‘2’, and ‘3’, it can be found that the model using only the edge feature augment block and the model using only the spectral attention block perform similarly on IP, KSC, and PU, while the model using only the spatial attention block does not perform as well. This is because for the models using either the edge feature augment block or the spectral attention block, this has the effect of removing noise, whereas using the spatial attention block directly introduces this invalid noise into the network, which adversely affects the model’s capabilities.

Comparing ‘4’, ‘5’, and ‘6’, the model performs similarly on all datasets at this point, and better than the model in ‘1’, ‘2’, and ‘3’. Comparing ‘3’ with ‘5’ and ‘6’, it can be found that the performance of the model gains a significant improvement, which greatly illustrates the need for edge feature augmentation and noise reduction processing of HSI using edge feature augment block and spectral attention block.

Overall, the edge feature augment block and the spectral attention block play a great role in suppressing the noise in HSIs, and combining them with the spatial attention block will result in better performance than other combinations.

## 5. Conclusions

In this paper, a novel feature extraction network (ESSN) is proposed for efficiently extracting local edge features and global spectral–spatial features from HSIs. In the ESSN, firstly, the edge feature augment block performs edge-aware and selective feature enhancement efficiently compared to the traditional edge data augmentation using the LoG operator with no learnable parameters. Secondly, due to the presence of a large amount of noise in some of the spectra in the HSI, different spectra do not have the same importance for the classification decision, so we introduce the spectral attention block to enhance the effective spectra and suppress the noise. Also, due to the geometric constraints of the convolutional operation, we introduce spatial attention to model the pixel–pixel interactions at all locations. Finally, we fuse representations of the feature maps reconstructed by the above methods through the 2D–3D convolution block to obtain the final feature representations. The experimental results show that ESSN performs competitively on the IP, KSC, and PU datasets.

Although ESSN has better performance in HSI classification, further improvements are needed. Afterwards, we will continue the following studies:Exploring better edge-aware algorithms so as to reduce noise interference from isolated nodes;Reduce the parameter size to speed up training and increase efficiency.

## Figures and Tables

**Figure 1 sensors-24-04714-f001:**
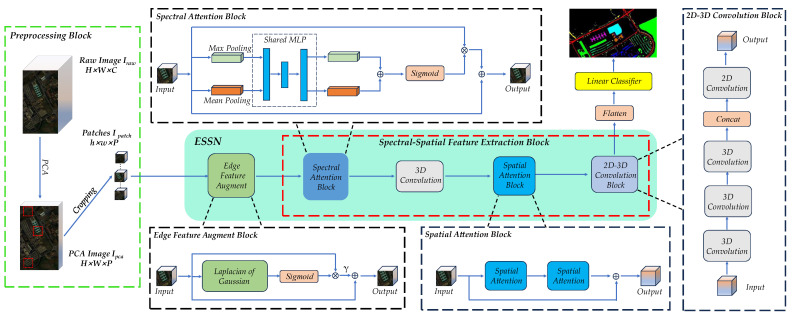
Framework of the classification process using ESSN. Note that BN and ReLU after each convolution operation have been omitted.

**Figure 2 sensors-24-04714-f002:**
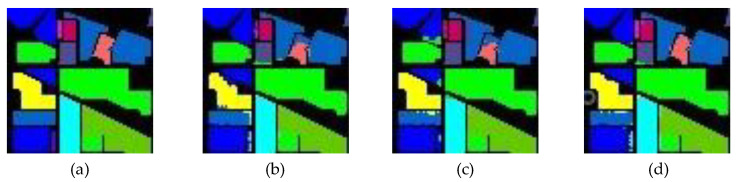
Comparison of prediction and truth plots. (**a**) Partial ground truth of IP, (**b**–**d**) Predicted classification maps.

**Figure 3 sensors-24-04714-f003:**
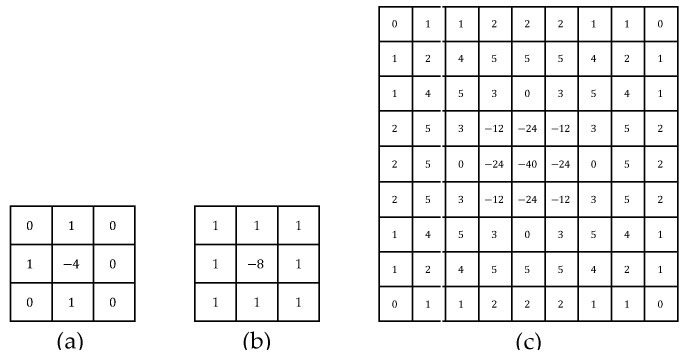
LoG operators for the two cases σ<0.5 and σ=1.4. (**a**,**b**) corresponding to σ<0.5, (**c**) corresponding to σ=1.

**Figure 4 sensors-24-04714-f004:**
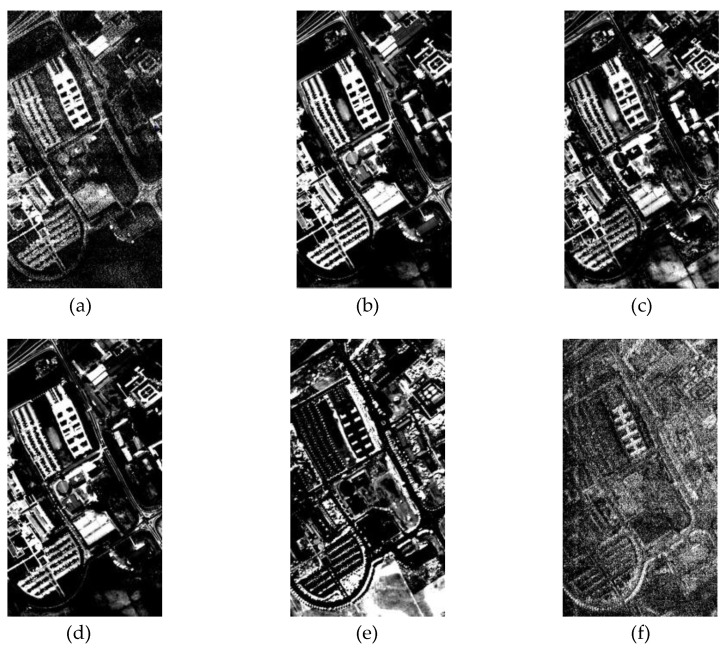
Binarized images of different spectral bands of PU dataset. (**a**–**c**) derived from raw PU dataset, (**d**–**f**) derived from PU dataset after PCA.

**Figure 5 sensors-24-04714-f005:**
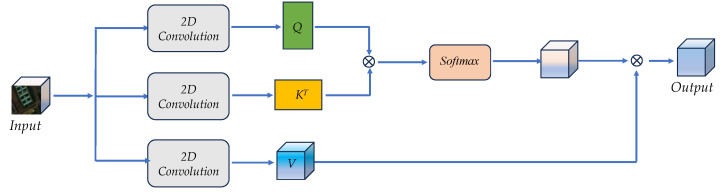
The framework of spatial attention in Figure 1.

**Figure 6 sensors-24-04714-f006:**
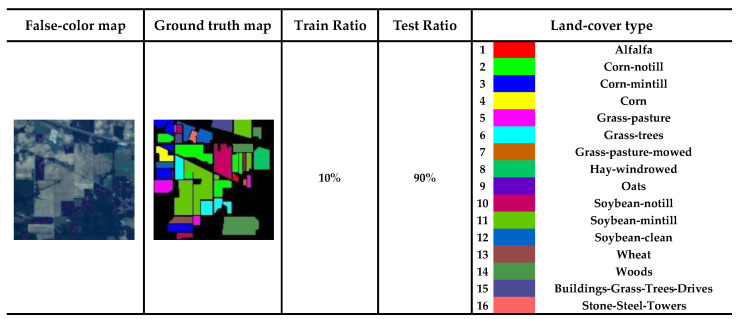
Specific information on the IP dataset.

**Figure 7 sensors-24-04714-f007:**
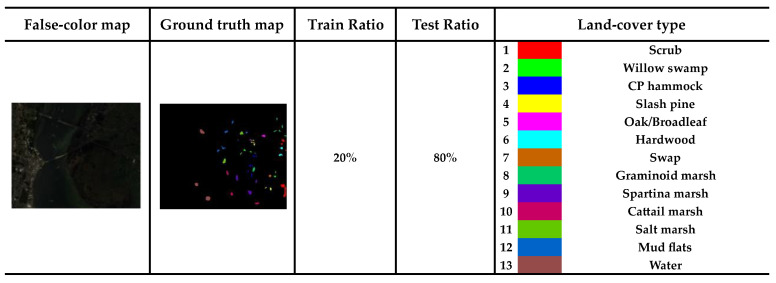
Specific information on the KSC dataset.

**Figure 8 sensors-24-04714-f008:**
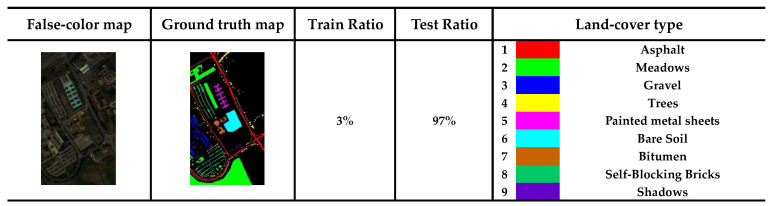
Specific information on the PU dataset.

**Figure 9 sensors-24-04714-f009:**
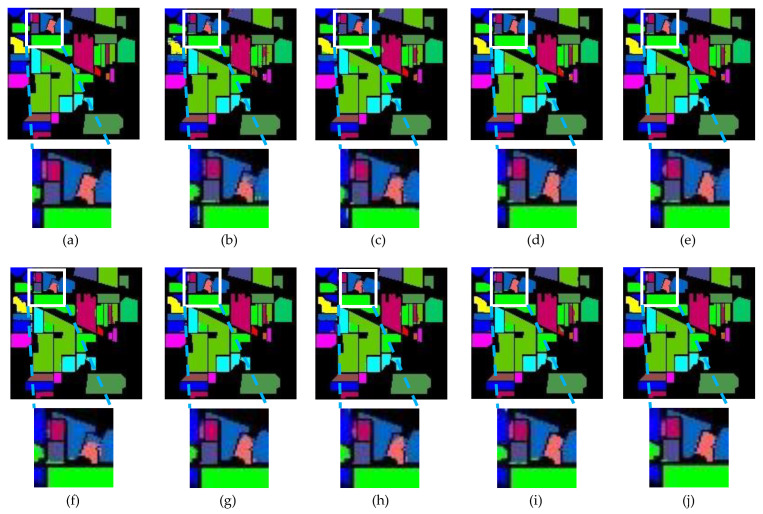
Performance on IP. (**a**) Full ground truth; (**b**) 2-D CNN (97.11%); (**c**) 3-D CNN (97.19%); (**d**) HybridSN (98.72%); (**e**) SSRN (98.61%); (**f**) SSTN (97.66%); (**g**) SSFTT (98.57%); (**h**) CTMixer (98.42%); (**i**) 3DCT (98.54%); (**j**) ESSN (98.93%).

**Figure 10 sensors-24-04714-f010:**
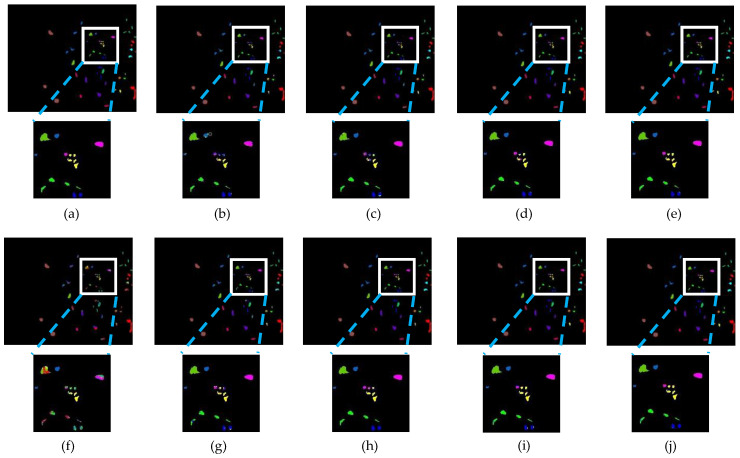
Performance on KSC. (**a**) Full ground truth; (**b**) 2-D CNN (91.35%); (**c**) 3-D CNN (93.04%); (**d**) HybridSN (98.99%); (**e**) SSRN (98.60%); (**f**) SSTN (63.16%); (**g**) SSFTT (92.35%); (**h**) CTMixer (95.14%); (**i**) 3DCT (97.87%); (**j**) ESSN (99.18%).

**Figure 11 sensors-24-04714-f011:**
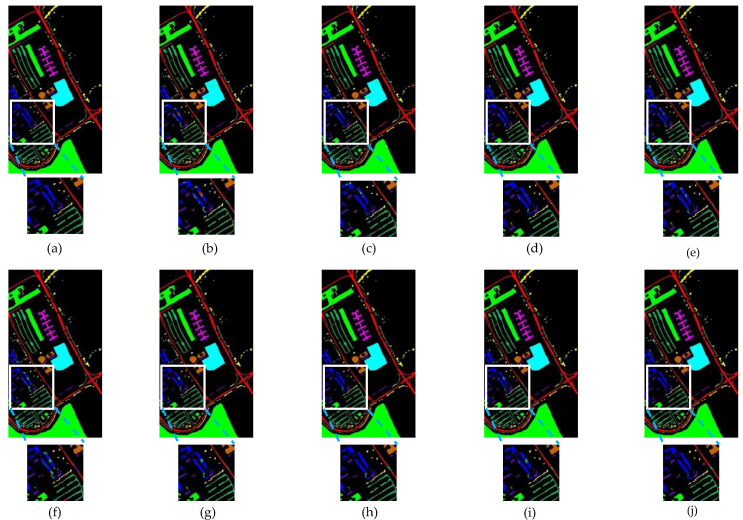
Performance on PU. (**a**) Full ground truth; (**b**) 2-D CNN (97.15%); (**c**) 3-D CNN (96.97%); (**d**) HybridSN (99.36%); (**e**) SSRN (99.12%); (**f**) SSTN (96.01%); (**g**) SSFTT (99.13%); (**h**) CTMixer (99.02%); (**i**) 3DCT (99.36%); (**j**) ESSN (99.45%).

**Figure 12 sensors-24-04714-f012:**
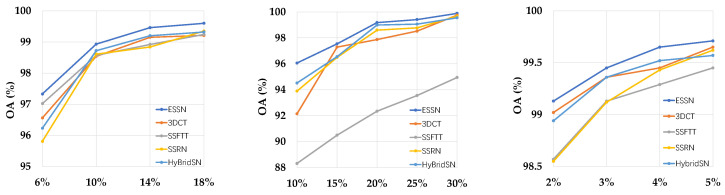
OA (%) at different training ratios on IP (**left**), KSC (**middle**), and PU (**right**), respectively.

**Figure 13 sensors-24-04714-f013:**
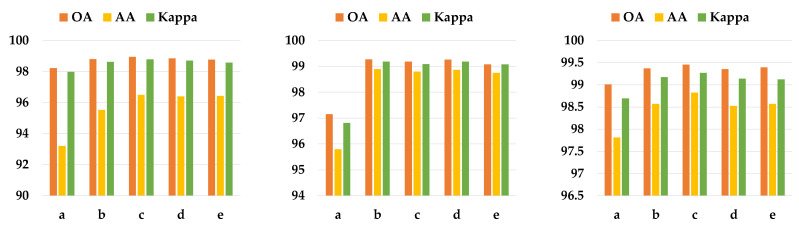
Classification accuracy (%) on IP (**left**), KSC (**middle**), and PU (**right**) in different cases. The ‘a’ denotes traditional edge data augmentation using the LoG operator with no learnable parameters. In turn, ‘b’, ‘c’, and ‘d’ correspond to the use of the (a), (b), and (c) operators in Figure 3, respectively. ‘e’ indicates no processing on edges, corresponding to case ‘6’ in Table 6.

**Figure 14 sensors-24-04714-f014:**
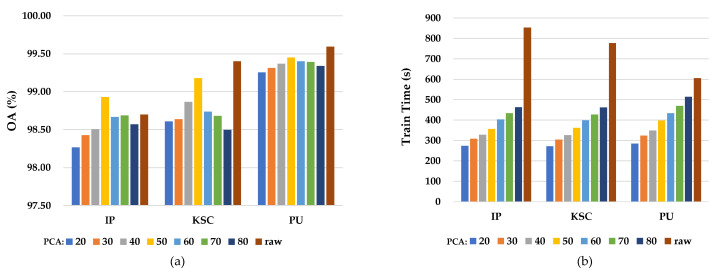
Model classification accuracy (subplot (**a**)) and train time cost (subplot (**b**)) after downscaling to different dimensions using PCA operations. ‘raw’ indicates that PCA is not used.

**Table 1 sensors-24-04714-t001:** Classification accuracy (%) of different models on IP.

Class	2-D CNN	3-D CNN	HybridSN	SSRN	SSTN	SSFTT	CTMixer	3DCT	ESSN
1	68.33 ± 22.14	88.60 ± 15.82	89.55 ± 13.29	82.06 ± 15.17	83.19 ± 20.67	90.92 ± 13.15	**92.86 ± 12.42**	84.95 ± 18.20	89.30 ± 12.81
2	96.29 ± 1.79	96.24 ± 1.30	98.32 ± 0.62	98.25 ± 0.52	97.75 ± 0.74	**98.53 ± 0.69**	98.03 ± 0.80	97.83 ± 0.76	98.39 ± 0.58
3	97.40 ± 0.94	95.81 ± 2.52	99.19 ± 0.87	99.02 ± 0.72	98.28 ± 1.35	98.80 ± 0.88	98.41 ± 1.00	98.56 ± 0.83	**99.34 ± 0.56**
4	92.37 ± 7.68	98.26 ± 2.55	98.77 ± 2.20	99.19 ± 1.53	97.86 ± 3.25	98.63 ± 2.22	**99.23 ± 1.65**	97.92 ± 2.56	98.86 ± 2.04
5	96.16 ± 1.76	97.86 ± 1.37	98.34 ± 1.21	98.16 ± 1.37	96.60 ± 1.85	98.27 ± 1.15	98.16 ± 1.22	98.29 ± 1.53	**98.80 ± 1.14**
6	99.27 ± 0.71	99.16 ± 0.68	99.39 ± 0.54	99.39 ± 0.43	98.52 ± 0.93	**99.50 ± 0.45**	99.25 ± 0.49	99.66 ± 0.24	99.48 ± 0.46
7	51.12 ± 28.95	92.69 ± 14.02	92.66 ± 12.10	92.66 ± 13.15	32.47 ± 37.73	92.20 ± 11.97	93.42 ± 7.14	92.68 ± 12.09	**97.29 ± 4.57**
8	**100.00 ± 0.00**	99.81 ± 0.39	**100.00 ± 0.00**	99.98 ± 0.07	99.95 ± 0.14	99.93 ± 0.21	99.98 ± 0.07	99.98 ± 0.07	99.98 ± 0.07
9	44.75 ± 35.70	63.97 ± 31.09	72.91 ± 37.31	71.08 ± 35.95	36.14 ± 40.49	65.35 ± 39.81	66.79 ± 37.46	72.89 ± 34.52	**73.31 ± 36.21**
10	96.60 ± 2.06	96.38 ± 1.33	98.96 ± 0.55	98.56 ± 0.65	97.52 ± 0.94	99.15 ± 0.75	98.09 ± 0.92	98.50 ± 0.63	**99.10 ± 0.55**
11	98.5 ± 0.69	98.16 ± 0.71	98.99 ± 0.50	99.00 ± 0.60	98.33 ± 0.59	98.84 ± 0.47	98.58 ± 0.70	98.75 ± 0.58	**99.14 ± 0.47**
12	93.41 ± 3.00	92.84 ± 3.36	96.11 ± 1.98	97.31 ± 1.47	94.38 ± 3.35	95.10 ± 2.60	95.98 ± 2.75	97.24 ± 1.73	**97.43 ± 1.62**
13	99.31 ± 0.66	97.89 ± 2.29	99.18 ± 1.96	99.38 ± 1.51	98.50 ± 3.17	**99.64 ± 0.65**	98.34 ± 3.17	99.22 ± 1.50	99.01 ± 1.60
14	99.72 ± 0.17	98.97 ± 0.35	**99.86 ± 0.14**	99.67 ± 0.30	99.71 ± 0.29	99.51 ± 0.50	99.78 ± 0.18	99.78 ± 0.29	99.78 ± 0.24
15	96.02 ± 3.50	96.13 ± 3.42	99.46 ± 0.99	97.49 ± 2.30	99.00 ± 1.10	98.51 ± 1.92	**99.54 ± 0.78**	98.56 ± 1.79	99.17 ± 0.92
16	89.27 ± 9.04	93.84 ± 3.25	91.22 ± 9.55	91.85 ± 4.78	87.85 ± 10.57	92.19 ± 4.65	93.17 ± 5.34	**95.50 ± 2.54**	95.49 ± 2.45
OA	97.11 ± 0.34	97.19 ± 0.60	98.72 ± 0.26	98.61 ± 0.24	97.66 ± 0.44	98.57 ± 0.32	98.42 ± 0.26	98.54 ± 0.30	**98.93 ± 0.17**
AA	88.91 ± 4.30	94.16 ± 3.71	95.81 ± 3.40	95.19 ± 3.33	88.50 ± 4.46	95.32 ± 3.31	95.60 ± 2.99	95.64 ± 3.52	**96.49 ± 3.14**
Kappa	96.71 ± 0.39	96.80 ± 0.68	98.54 ± 0.30	98.41 ± 0.28	97.33 ± 0.51	98.37 ± 0.36	98.20 ± 0.30	98.33 ± 0.36	**98.78 ± 0.19**

**Table 2 sensors-24-04714-t002:** Classification accuracy (%) of different models on KSC.

Class	2-D CNN	3-D CNN	HybridSN	SSRN	SSTN	SSFTT	CTMixer	3DCT	ESSN
1	97.76 ± 1.45	98.87 ± 0.92	99.88 ± 0.21	**100.00 ± 0.00**	81.15 ± 28.39	97.73 ± 1.76	89.94 ± 29.92	99.82 ± 0.29	99.90 ± 0.21
2	68.97 ± 8.81	81.87 ± 7.90	97.22 ± 2.75	96.16 ± 2.32	6.46 ± 10.68	86.63 ± 7.12	90.07 ± 15.79	94.28 ± 4.99	**97.99 ± 2.17**
3	90.74 ± 8.56	86.23 ± 6.42	**95.43 ± 2.70**	93.43 ± 2.00	0.15 ± 0.33	82.29 ± 8.75	90.27 ± 20.86	93.36 ± 3.78	95.30 ± 3.06
4	69.68 ± 5.98	74.79 ± 8.12	94.14 ± 3.50	90.06 ± 3.50	17.25 ± 16.60	62.24 ± 13.25	87.87 ± 17.11	91.24 ± 3.83	**94.89 ± 3.43**
5	82.52 ± 5.14	90.06 ± 2.65	97.86 ± 2.05	97.25 ± 2.42	30.86 ± 21.51	90.96 ± 4.61	93.22 ± 8.70	97.28 ± 2.45	**98.76 ± 0.88**
6	78.21 ± 13.17	85.85 ± 4.35	98.79 ± 1.05	96.46 ± 3.01	10.88 ± 8.68	85.79 ± 10.39	82.66 ± 24.62	95.40 ± 5.79	**99.13 ± 1.04**
7	86.47 ± 10.44	94.57 ± 5.55	99.28 ± 1.10	**100.00 ± 0.00**	0.00 ± 0.00	95.18 ± 11.85	88.92 ± 29.72	96.29 ± 6.00	**100.00 ± 0.00**
8	94.26 ± 3.67	92.27 ± 3.24	99.65 ± 0.45	99.53 ± 0.60	73.43 ± 18.12	91.23 ± 9.51	95.73 ± 5.44	98.53 ± 1.82	**99.62 ± 0.41**
9	96.36 ± 4.63	88.58 ± 3.51	99.43 ± 1.06	99.20 ± 1.57	54.91 ± 29.11	88.89 ± 5.78	98.56 ± 2.53	97.67 ± 2.36	**99.55 ± 0.89**
10	88.23 ± 3.64	93.14 ± 3.55	98.72 ± 1.87	99.88 ± 0.28	73.91 ± 19.82	95.33 ± 3.05	98.50 ± 2.01	97.25 ± 1.56	**99.31 ± 1.22**
11	98.13 ± 1.42	97.44 ± 1.71	**100.00 ± 0.00**	**100.00 ± 0.00**	89.62 ± 9.54	98.44 ± 1.27	99.85 ± 0.36	**100.00 ± 0.00**	**100.00 ± 0.00**
12	87.56 ± 2.84	95.48 ± 1.54	99.73 ± 0.44	99.71 ± 0.24	72.68 ± 10.91	93.71 ± 5.25	99.73 ± 0.36	98.67 ± 1.18	**99.83 ± 0.31**
13	99.93 ± 0.12	99.88 ± 0.16	**100.00 ± 0.00**	**100.00 ± 0.00**	97.82 ± 5.43	**100.00 ± 0.00**	99.99 ± 0.04	**100.00 ± 0.00**	**100.00 ± 0.00**
OA	91.35 ± 1.22	93.04 ± 0.98	98.99 ± 0.30	98.60 ± 0.34	63.16 ± 6.48	92.35 ± 92.35	95.14 ± 9.21	97.87 ± 0.56	**99.18 ± 0.32**
AA	87.60 ± 1.83	90.69 ± 1.46	98.47 ± 0.49	97.82 ± 0.48	46.85 ± 6.65	89.88 ± 1.89	93.49 ± 11.31	96.91 ± 0.99	**98.79 ± 0.49**
Kappa	90.36 ± 1.36	92.25 ± 1.10	98.87 ± 0.34	98.44 ± 0.38	58.36 ± 7.49	91.48 ± 1.22	94.61 ± 10.20	97.61 ± 0.62	**99.08 ± 0.36**

**Table 3 sensors-24-04714-t003:** Classification accuracy (%) of different models on PU.

Class	2-D CNN	3-D CNN	HybridSN	SSRN	SSTN	SSFTT	CTMixer	3DCT	ESSN
1	96.44 ± 1.08	96.12 ± 0.88	**99.85 ± 0.22**	99.62 ± 0.26	95.23 ± 4.78	99.70 ± 0.18	99.59 ± 0.35	99.63 ± 0.27	99.76 ± 0.35
2	**99.99 ± 0.01**	99.79 ± 0.12	**99.99 ± 0.01**	**99.99 ± 0.01**	99.82 ± 0.20	**99.99 ± 0.01**	99.98 ± 0.04	99.95 ± 0.04	99.99 ± 0.03
3	88.773 ± 3.30	88.99 ± 3.73	95.79 ± 2.50	92.99 ± 2.91	79.55 ± 11.40	96.49 ± 2.24	**97.87 ± 1.63**	94.83 ± 2.46	97.38 ± 1.69
4	93.36 ± 1.52	92.46 ± 2.20	96.95 ± 1.09	97.32 ± 0.44	91.64 ± 3.41	95.34 ± 1.40	94.02 ± 1.99	**98.32 ± 0.67**	97.39 ± 0.86
5	99.95 ± 0.09	99.97 ± 0.70	99.99 ± 0.05	99.82 ± 0.25	96.35 ± 3.86	99.86 ± 0.17	99.91 ± 0.11	**100.00 ± 0.00**	99.98 ± 0.05
6	98.37 ± 0.93	99.24 ± 0.78	99.99 ± 0.04	99.97 ± 0.10	96.85 ± 8.47	**100.00 ± 0.00**	99.84 ± 0.47	99.86 ± 0.22	**100.00 ± 0.00**
7	98.17 ± 2.02	99.50 ± 0.66	**99.99 ± 0.02**	99.53 ± 0.62	88.32 ± 15.45	99.73 ± 0.55	99.92 ± 0.21	99.86 ± 0.21	99.97 ± 0.09
8	89.91 ± 2.98	90.51 ± 1.78	**99.11 ± 0.73**	97.82 ± 0.76	95.70 ± 3.84	97.64 ± 1.15	98.15 ± 1.15	98.62 ± 0.73	98.78 ± 0.67
9	93.44 ± 3.81	85.33 ± 3.12	95.07 ± 2.57	96.62 ± 2.01	83.98 ± 9.21	95.63 ± 1.63	91.68 ± 2.23	**97.93 ± 1.35**	96.16 ± 1.51
OA	97.15 ± 0.31	96.97 ± 0.40	99.36 ± 0.12	99.12 ± 0.15	96.01 ± 1.93	99.13 ± 0.15	99.02 ± 0.24	99.36 ± 0.18	**99.45 ± 0.10**
AA	95.38 ± 0.54	94.66 ± 0.73	98.52 ± 0.33	98.19 ± 0.27	91.94 ± 3.53	98.27 ± 0.35	97.88 ± 0.43	98.78 ± 0.36	**98.82 ± 0.24**
Kappa	96.22 ± 0.41	95.99 ± 0.53	99.15 ± 0.16	98.83 ± 0.20	94.68 ± 2.60	98.85 ± 0.19	98.71 ± 0.32	99.15 ± 0.24	**99.27 ± 0.13**

**Table 4 sensors-24-04714-t004:** Resource consumption for each model.

Method	Params (M)	IP		KSC		PU	
		Train Time (s)	Test Time (s)	Train Time (s)	Test Time (s)	Train Time (s)	Test Time (s)
2-D CNN	0.04	200.85	2.54	197.80	2.12	199.25	5.64
3-D CNN	0.62	253.21	4.03	255.09	2.64	268.24	10.52
HybridSN	4.34	319.42	4.93	319.78	3.16	337.16	15.85
SSRN	0.47	291.30	4.56	293.64	3.07	308.85	14.61
SSTN	0.02	228.26	3.08	227.89	2.36	236.12	8-17
SSFTT	0.23	207.70	2.54	205.24	2.19	208.70	5.86
CTMixer	0.60	262.06	3.96	267.03	2.70	280.81	12.20
3DCT	3.84	346.95	6.19	349.57	3.77	384.90	22.05
ESSN	4.37	332.87	5.24	330.77	3.36	354.41	17.32

**Table 5 sensors-24-04714-t005:** OA (%) on different patch sizes.

Patch Size	11 × 11	13 × 13	15 × 15	17 × 17	19 × 19
IP	98.62 ± 0.31	98.72 ± 0.24	**98.93 ± 0.17**	98.76 ± 0.28	98.73 ± 0.25
KSC	98.60 ± 0.39	98.93 ± 0.44	99.18 ± 0.32	99.37 ± 0.20	**99.58 ± 0.20**
PU	99.18 ± 0.10	99.37 ± 0.16	**99.45 ± 0.10**	99.42 ± 0.16	99.37 ± 0.16

**Table 6 sensors-24-04714-t006:** Classification accuracy (%) of different combinations of block.

Case	Edge Block	Spectral Block	Spatial Block	Metric	IP	KSC	PU
1	√	×	×	OA	98.74 ± 0.24	99.00 ± 0.29	99.39 ± 0.11
				AA	95.72 ± 3.01	98.52 ± 0.48	98.79 ± 0.32
				Kappa	98.68 ± 0.27	98.88 ± 0.32	99.21 ± 0.14
2	×	√	×	OA	98.75 ± 0.25	99.06 ± 0.32	99.38 ± 0.08
				AA	96.02 ± 3.28	98.77 ± 0.45	98.59 ± 0.24
				Kappa	98.58 ± 0.29	99.06 ± 0.36	99.11 ± 0.10
3	×	×	√	OA	76.25 ± 8.97	55.88 ± 13.02	89.70 ± 4.60
				AA	70.23 ± 11.22	46.93 ± 12.72	83.67 ± 8.98
				Kappa	73.08 ± 9.98	50.30 ± 14.54	86.33 ± 6.09
4	√	√	×	OA	98.82 ± 0.31	99.12 ± 0.18	99.40 ± 0.14
				AA	95.86 ± 3.41	98.72 ± 0.28	98.65 ± 0.44
				Kappa	98.66 ± 0.35	99.03 ± 0.20	99.21 ± 0.18
5	√	×	√	OA	98.77 ± 0.20	99.16 ± 0.23	99.40 ± 0.15
				AA	95.95 ± 3.55	98.75 ± 0.35	98.61 ± 0.43
				Kappa	98.59 ± 0.23	99.04 ± 0.26	99.17 ± 0.20
6	×	√	√	OA	98.77 ± 0.27	99.07 ± 0.24	99.39 ± 0.14
				AA	96.43 ± 3.25	98.75 ± 0.32	98.57 ± 0.29
				Kappa	98.57 ± 0.30	99.07 ± 0.27	99.12 ± 0.19
7	√	√	√	**OA**	**98.93 ± 0.17**	**99.18 ± 0.32**	**99.45 ± 0.10**
				**AA**	**96.49 ± 3.14**	**98.79 ± 0.49**	**98.82 ± 0.24**
				**Kappa**	**98.78 ± 0.19**	**99.08 ± 0.36**	**99.27 ± 0.13**

## Data Availability

Data are contained within the article.

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
