# Peer review of "A Joint Network of Edge-Aware and Spectral–Spatial Feature Learning for Hyperspectral Image Classification"

_sensors, 2024, doi:10.3390/s24144714_

Round 1

Reviewer 1 Report

Comments and Suggestions for Authors

In this paper, the authors propose a joint network of edge-aware and spectral-spatial feature learning for hyperspectral image classification. This work focuses on feature extraction of the spectral, spatial, and edge-attention mechanism for hyperspectral images and finally enhancement of image spectral-spatial features using a 2D-3D convolution module. Although the authors validated the classification performance on three hyperspectral images and proved the effectiveness of the method, there are still some concerns with this paper.

1. In addition to describing the advantages and disadvantages of HSI and RGB images in the INTRODUCTION section, the authors should point out the imaging link between HSI and RGB, e.g., doi:10.1109/TGRS.2024.3361929.

2. the authors have been emphasizing that image edge information enhancement can overcome the misclassification phenomenon of two-class boundaries, but it is not clear how the introduction of such edge information improves the classification accuracy of two-class boundaries.

3. spectral feature extraction of raw HSI using PCA before performing the classification task, did the authors consider the loss of spectral information, explicit experiments are needed.

4. 3D convolution is a combination of 1D and 2D convolution considered from spectral and spatial level. So, what is the authors' motivation for mixing 2D and 3D convolution?

5. On the PU dataset, only one class of feature accuracy achieved the highest accuracy, but the OA, AA, and Kappa metrics were the highest, and the authors should explain in detail why.

6. The parameter model parameter number to see, the author method is the largest of all the methods in the parameter number of one, and the author uses the parameter number of more small method author competition method, which seems to be unfair, it is recommended that the authors of at least 2 of the same parameter quantitative level of the introduction of the method for a fair comparison.

7. Two 'spatial attention' appear in the header of Table 6, and the authors are asked to scrutinize them.

8. It is recommended that the authors add the limitation of methods in the CONCLUSION section.

Comments on the Quality of English Language

Moderate editing of the English language required

Reviewer 2 Report

Comments and Suggestions for Authors

The task of hyperspectral image classification is considered. A pipelined processing is used, some blocks of the pipeline have original architecture proposed by the authors. The work contains novelty and is grounded by experimental study. It can be published as is.

Author Response

Thank you very much for the reviewers' recognition of our work.

Reviewer 3 Report

Comments and Suggestions for Authors This paper focuses on classifying hyperspectral images through the development of edge extraction and spatial-spectral feature extraction modules. To enhance the quality of the paper, please consider the following comments:   1. The motivations presented are unclear and highly confusing. The introduction section should be rewritten to logically and technically clarify the motivations.   2. The paper is difficult to read and comprehend. The quality of the English writing needs significant improvement.   3. The authors mention previous methods of edge data augmentation but do not specify what these methods are. What distinguishes the proposed method from the existing ones?   4. In the experimental section, the results of the proposed method on the UP dataset are not competitive. Can the authors elucidate why this might be the case?   5. The discussion should include more spatial-spectral feature learning methods, such as those outlined in the following references: 10.1109/TGRS.2022.3159789, 10.1109/LGRS.2021.3070074, 10.1109/TGRS.2022.3153673, and 10.1109/LGRS.2022.3200145. Comments on the Quality of English Language

This paper needs thorough revisions, addressing both the technical quality and the clarity of the English presentation.

Round 2

Reviewer 1 Report

Comments and Suggestions for Authors

The author has solved all my problems.

Comments on the Quality of English Language

The author has solved all my problems.

Reviewer 3 Report

Comments and Suggestions for Authors

The authors have responded to all the concerns raised by the reviewer. The manuscript is deemed suitable for publication.